# Various facets of intermolecular transfer of phase coherence by nuclear dipolar fields

Philippe Pelupessy[1]

[1]Laboratoire des Biomolécules, LBM, Département de Chimie, École Normale Supérieure, PSL University, Sorbonne Université, CNRS, 75005 Paris, France

**Correspondence:** Philippe Pelupessy (philippe.pelupessy@ens.psl.eu)

**Abstract.** It has long been recognized that dipolar fields can mediate intermolecular transfer of phase coherence from abundant solvent to sparse solute spins. Generally, the dipolar field has been considered while acting during prolonged free precession delays. Recently, we have shown that transfer can also occur during suitable uninterrupted radio-frequency pulse-trains that are used for total correlation spectroscopy. Here, we will expand upon the latter work. First, analytical expressions for the evolution of the solvent magnetization under continuous irradiation and the influence of the dipolar field are derived. These expressions facilitate the simulations of the transfer process. Then, a pulse sequence for the retrieval of high-resolution spectra in inhomogeneous magnetic fields is described, and another sequence to detect a transfer from an intermolecular double quantum coherence. Finally, various schemes are discussed where the magnetization is modulated by a combination of multiple selective radio frequency pulses and pulsed field gradients along different directions. In these schemes, the magnetization is manipulated in such a way that the dipolar field, which originates from a single spin species, can be decomposed into two components. Each component originates from a part of the magnetization that is modulated in a different direction. Both can independently, but simultaneously, mediate an intermolecular transfer of phase coherence.

## 1 Introduction

In liquid state NMR, the magnetization of an abundant or a highly polarized spin species affects the evolution of the density operator through radiation damping (RD) (Suryan, 1949) and through the dipolar field (DF) (Dickinson, 1951). RD stems from the radio-frequency (rf) field caused by the current that the transverse magnetization induces in an rf coil (Bloembergen and Pound, 1954). The DF describes the direct contributions of the longitudinal and transverse nuclear magnetization components to the static field $B_0$ and to a perpendicular rf field $B_1$, respectively, through dipolar interactions. It is also known as the DDF which in older works stood for the *dipolar demagnetizing field* (Deville et al., 1979) but which in more recent works has been redefined as the *distant dipolar field* (Ahn et al., 1998). Both RD and the DF can be incorporated into a modified set of Bloch equations (Bloom, 1957; Deville et al., 1979). Since RD results in a field with only a rapid oscillating transverse component, its effect on other resonances is usually limited to nearby frequencies. When the chemical shift differences are removed from the effective Hamiltonian by suitable pulse sequences, the effects of RD extend over a much wider range of frequencies (Pelupessy, 2022a). Conversely, the DF has a longitudinal component which causes a shift in the precession

frequencies of all nuclei that possess a spin (Edzes, 1990). Striking effects are observed when the magnetization of an abundant spin species depends on its spatial position, often as a result of a pulsed field gradient (PFG). These non-trivial effects include multiple spin echoes in two-pulse experiments (Bernier and Delrieu, 1977; Bowtell et al., 1990) and intermolecular multiple quantum cross-peaks in COSY-like (correlation spectroscopy) sequences. These peaks can stem from $like$ (He et al., 1993) or $unlike$ (Warren et al., 1993) spins. The phase-coherence of abundant spins can also be transferred by the DF during pulse-

trains that are commonly used in homo-nuclear total correlation spectroscopy (TOCSY) (Pelupessy, 2022b). As with RD in these type of experiments, the small transverse component of the DF plays an important role even if chemical shift differences are large.

  In this work, several aspects of the transfer that is mediated by the DF and occurs during continuous pulse-trains will be explored: broadband in-phase transfer and high-resolution spectra can be obtained in inhomogeneous $B_0$ fields in a fashion

similar to the HOMOGENIZED (homogeneity enhancement by intermolecular zero-quantum detection, Vathyam et al. (1996)) and related experiments (Lin et al., 2013). In addition, a transfer of phase coherence can be realized from intermolecular double quantum (DQ) coherences involving the abundant solvent and the sparse solute spins. Finally, experiments where the magnetization is modulated in more complex ways by applying several PFGs in combination with selective rf pulses will be discussed.

## 2 Theory

### 2.1 The evolution of the magnetization during rf pulse-trains

The following theory, originally developed by Deville et al. (1979), applies to a homo-nuclear spin system that contains an abundant spin species $A$ and a sparse spin species $S$ (both having a spin of $1/2$), where the magnetization of the spins $A$ has been modulated in a manner that it averages out over the effective sample volume (Warren et al. (1995) expanded the theory for

the case where this condition is not met). Moreover, the spatial variations must be in a single direction $\mathbf{s}$. These modulations are usually induced by a field gradient which is oriented at an angle $\theta_G$ with respect to $B_0$, by convention along the $z$-axis, so that $\cos\theta_G = \hat{s} \cdot \hat{z}$. The DF can then be characterized by an angular frequency $\omega_d$ defined as:

$$\omega_d = \frac{1}{3}\mu_0\gamma M_{eq}^A(3\cos^2\theta_G - 1) \ , \tag{1}$$

with $\gamma$ the gyro-magnetic ratio of the $A$ and $S$ spins, $\mu_0$ the vacuum permeability, and $M_{eq}^A$ the magnitude of the magnetization

of the $A$ spins at equilibrium. In the rotating frame, the evolution of the magnetization vectors of both $A$ and $S$ spins is governed by the modified Bloch equations (Deville et al., 1979; Bowtell et al., 1990; Enss et al., 1999):

$$\begin{aligned}
\dot{m}_x^i &= & -(\omega_0^i + \omega_d m_z^A)m_y^i &+ (\omega_{1y} - \omega_d m_y^A/2)m_z^i \ , \\
\dot{m}_y^i &= & (\omega_0^i + \omega_d m_z^A)m_x^i &- (\omega_{1x} - \omega_d m_x^A/2)m_z^i \ , \\
\dot{m}_z^i &= & -(\omega_{1y} - \omega_d m_y^A/2)m_x^i &+ (\omega_{1x} - \omega_d m_x^A/2)m_y^i \ ,
\end{aligned} \tag{2}$$

with $i$ either $A$ or $S$, $\omega_{1x/y}$ the (time-dependent) rf field, and $\omega_0$ the offset from the carrier frequency. The magnetization components, written in lower case $m$, are normalized with respect to the equilibrium amplitudes. These equations are local (as a result of the modulations being only in one direction Deville et al. (1979)) so that the evolution can be calculated separately for each position in the sample. Typically, the rf term does not appear in these equations, since previously the DF has been considered predominantly during free precession delays. The explicit position and time dependence of the variables has been omitted in these equations. A set time $t = T$ will be indicated in brackets. Neither relaxation nor molecular motions by diffusion or convection have been taken into account.

The set of coupled eqs. 2 is non-linear when $i = A$, while for the sparse spins $i = S$ the magnetization of the $A$ spins $\mathbf{m}^A = (m_x^A, m_y^A, m_z^A)$ is the source of a time-dependent field $\gamma B_d = (-\omega_d m_x^A/2, -\omega_d m_y^A/2, \omega_d m_z^A)$. Hence, the evolution of the magnetization of the $S$ spins $\mathbf{m}^S = (m_x^S, m_y^S, m_z^S)$ can be calculated straightforwardly from the trajectory of $\mathbf{m}^A$. In Pelupessy (2022b), the evolution of $\mathbf{m}^A$ during the rf pulse-trains has been obtained by numerical integration of the non-linear set of eqs. 2. In this work, a DIPSI-2 (decoupling in presence of scalar interactions, Rucker and Shaka (1989)) pulse-train is always applied at the resonance frequency of the $A$-spins, in which case the trajectory of $\mathbf{m}^A$ can be approximated analytically as follows: in appendix A, it is derived that for a strong constant on-resonance rf field ($|\omega_1| \gg |\omega_d|$) along the $x$-axis, $\mathbf{m}^A$ rotates around $x$ with an angular frequency of $\omega_{1x} - 3m_x^A(0)\omega_d/4$:

$$
\begin{aligned}
m_x^A &= m_x(0), \\
m_y^A &= \cos\left\{\omega_{1x}t - 3\omega_d m_x^A(0)t/4\right\} m_y^A(0) - \sin\left\{\omega_{1x}t - 3\omega_d m_x^A(0)t/4\right\} m_z^A(0), \\
m_z^A &= \cos\left\{\omega_{1x}t - 3\omega_d m_x^A(0)t/4\right\} m_z^A(0) + \sin\left\{\omega_{1x}t - 3\omega_d m_x^A(0)t/4\right\} m_y^A(0).
\end{aligned} \tag{3}
$$

Many sequences used for decoupling or magnetization transfer consist of a repetitive cycle of phase-alternated pulses along one axis (in this work, assumed between $+x$ and $-x$), as for example DIPSI-2, Waltz-16 (Shaka et al., 1983) and GARP (globally optimized alternating-phase rectangular pulses, Shaka et al. (1985)). Typically, the pulses are constant in amplitude, but may differ in duration. By design, the integral of the rf field $\int \omega_{1x} dt$ averages to zero over one cycle. Consequently, the contribution of $\omega_{1x}$ vanishes after each full cycle if eqs. 3 govern the evolution during all pulses in the cycle. In fig. 1a-c, the validity of this approximation is tested for DIPSI-2 by comparing the trajectories predicted by eqs. 3 with exact numerical simulations as described in Pelupessy (2022b). When the initial magnetization is oriented far away from the $x$-axis, the trajectories diverge at longer irradiation times $t_p$ of the DIPSI-2 pulse-train. These differences stem from small contributions of the oscillating terms in eqs. A5, due to the rapid switching of rf phases, which accumulate as the number of cycles increases. For shorter times $t_p < 0.4$ s, the curves obtained by both methods agree quite well.

For a continuous unmodulated rf field, the trajectories of $\mathbf{m}^A$ calculated with the two methods are indistinguishable, while for a GARP pulse-train, where the phases are alternated 2.5 times more frequently than for DIPSI-2 with the same rf amplitude, the deviations are much larger (see supporting information). Although one might be tempted to attribute this result to the higher phase switching rate, the details of the sequence are also important. For example, the use of WALTZ-16, with a phase switching rate that is 1.6 times more frequent than for DIPSI-2, results in a perfect match between the trajectories calculated with the two

methods. The elementary block of WALTZ-16 consists of three pulses $90^\circ_x 180^\circ_{-x} 270^\circ_x$, i.e. a $180^\circ$ pulse flanked by two pulses whose sum is $360^\circ$. These angles make the oscillating terms in eqs. A5 run over exactly 1 and 2 full rotations.

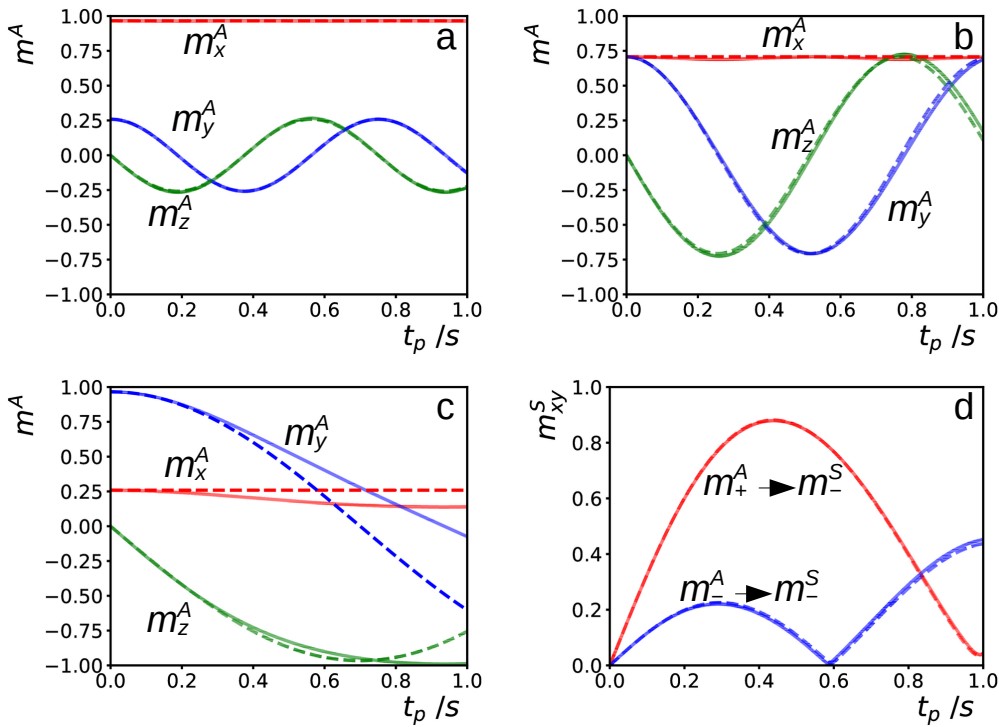

**Figure 1. (a-c)** Simulations of the evolution of the magnetization of the abundant spins $A$, during a time $t_p$ of an on-resonance DIPSI-2 pulse-train, for different initial conditions corresponding to magnetization in the $xy$-plane at angles of $15^\circ$(a), $45^\circ$(b) and $75^\circ$(c) with respect to the $x$-axis. The amplitude of the rf field was $\omega_1/2\pi$ = 8.33 kHz and the DF was characterized by $\omega_d = 2\pi \times 1.84$ rad s$^{-1}$. The magnetization is recorded stroboscopically after each full cycle. The solid lines result from numerical integration of the modified Bloch equations 2 as in Pelupessy (2022b). The dashed lines correspond to eqs. 3 with $\omega_{1x} = 0$. In the supporting information, similar simulations with GARP irradiation are shown. For continuous unmodulated irradiation or a WALTZ-16 pulse-train, no significant differences appear between the two different ways of calculating the evolution of the magnetization (i.e., the results of either method follow the dashed curves in this figure). **(d)** Simulation of the transfer of phase coherence from spins $A$ to $S$ with the experiment described in Pelupessy (2022b): a selective excitation of the $A$ spins followed by a DIPSI-2 pulse-train bracketed by two PFGs of equal area which have either equal (to observe a transfer $m_+^A \rightarrow m_-^S$) or opposite signs ($m_-^A \rightarrow m_-^S$). The parameters for the simulation were the same as in (a-c). The difference in chemical shift between the $A$ and $S$ spins was set to -3806 Hz. Similarly to (a-c), the trajectory of the magnetization of the $A$ spins at each position has been obtained either by numerical integration (leading to the solid lines) or by neglecting the DF within each DIPSI-2 cycle while using the approximate solution of eqs. 3 for the global evolution between the cycles (dashed lines). The gradients are assumed to be linear.

Fig. 1d shows simulations of the intermolecular transfer of phase coherence due to the DF from spins $A$ to spins $S$ for a gradient-selected selective TOCSY experiment (Dalvit and Bovermann, 1995; Pelupessy, 2022b). As in fig. 1a-c, the solid lines have been obtained with the trajectory of $\mathbf{m}^A$ calculated by numerical integration of the non-linear coupled differential eqs. 2 as described in Pelupessy (2022b), while the dashed lines with the trajectory predicted by eqs. 3. For the latter simulations (see the supporting information for the code), eqs. 3 with $\omega_{1x} = 0$ have been used to calculate the evolution of $\mathbf{m}^A$ between the DIPSI-2 cycles, while within each cycle the trajectory was assumed to be solely determined by the rf field. The dashed lines that can be calculated very rapidly are barely distinguishable from the solid ones that require the more laborious simulations of Pelupessy (2022b).

Effects of RD may mask or dampen those of the DF since the timescale in which RD occurs can be more than an order of magnitude shorter (Desvaux, 2013). However, in the experiments shown in this work, the former are suppressed by de-phasing the $A$ spins with a $90°$ pulse followed by a PFG, which allows us to focus on effects of the DF.

## 2.2 A qualitative description of the experiments

In the next section, several TOCSY-like experiments, where the DF mediates an intermolecular transfer of phase coherence, will be discussed. While the above theory will be used to simulate the transfer, the formalism developed by Warren and coworkers (Lee et al., 1996) will be employed to guide the experimental design. For this, the high temperature approximation needs to be abandoned by taking into account higher order terms in the expansion of the density matrix:

$$\rho_{eq} = (\mathbb{1} - S_z) \prod_i (\mathbb{1} - A_z^i) . \tag{4}$$

Since intermolecular interactions between sparse spins $S$ can be neglected, the index $i$ runs over all abundant spins $A$. In the experiments explored in this work, the equilibrium density operator evolves during a preparation period – before the DIPSI-2 irradiation – under the influence of rf pulses and PFGs. In this period, the DF will be neglected either because of its brevity or because the relevant part of the density operator commutes with the effective dipolar Hamiltonian. The intermolecular dipolar interactions between the $A$ and $S$ spins need to be tracked. The effective intermolecular dipolar Hamiltonian during the DIPSI-2 sequence is given by (Kramer et al., 2001):

$$H_{AS}^{eff} = -A_x S_x + (A_y S_y + A_z S_z)/2 . \tag{5}$$

The transfer induced by the DF can be evaluated by calculating the commutator of the density matrix and this Hamiltonian. The precise coefficients have been omitted in these equations since they are not needed for a mere qualitative description. Likewise, only a minimal number of spins $A$ will be considered that may give an observable signal on the $S$ spins. In this work, it is sufficient to account for two-spin operators containing one $A$ and one $S$ spin, since the different transfers involve only operators containing single quantum (SQ) terms of the $A$ spins (the experiment described in section 3.2 involves a DQ coherence which is a product of SQ terms of the $A$ and $S$ spins). For a qualitative description of the experiments, SQ operators containing only a single $A$ spin are sufficient. For quantitative calculations, also SQ operators containing multiple $A$ spins (such as $A_+^1 A_-^2 A_+^3$) are needed (Lee et al., 1996). The $A$ spins are assumed to have the same spatial coordinates as the $S$

spins. These simplifications preclude a precise description of the evolution due to the DF, neither the correct amplitude nor the angular dependence can be predicted. Nevertheless, it allowed a good insight into the original experiment where a DIPSI-2 sequence was used to transfer the coherence between $A$ and $S$ spins, and even provided a close estimate of the ratio of the initial rates of transfer into the different coherence orders (Pelupessy, 2022b).

## 3 Experiments

In Pelupessy (2022b), we demonstrated that the DF can efficiently mediate an intermolecular transfer of phase coherence during DIPSI-2 pulse-trains. In anterior work, where the DF acted during free precession delays, the change of coherence order needed to be achieved by rf pulses. On the contrary, the effective dipolar Hamiltonian during the DIPSI-2 irradiation allows for a change in coherence order by the DF. The transfer from a $+1$ to a $-1$ coherence was shown to be particularly efficient. This coherence order pathway refocuses $B_0$ inhomogeneities. In section 3.1, the original experiment is adapted to obtain high-resolution spectra in inhomogeneous fields in a HOMOGENIZED-like (Vathyam et al., 1996) fashion. In section 3.2, an alternative coherence selection pathway will be investigated; the transfer mediated by the DF from an intermolecular DQ coherence to $z$ magnetization. When multiple PFGs in different directions combined with several rf pulses are applied, the modulation pattern of the magnetization can become more complex. In section 3.3, the influence of this kind of modulations on the intermolecular transfer will be investigated.

### 3.1 High-resolution spectra in inhomogeneous fields

Fig. 2a depicts an adaptation of the selective TOCSY pulse sequence (Pelupessy, 2022b) with DIPSI-2 rf irradiation used to transfer the phase coherence from the abundant solvent spins $A$ to the solute spins $S$, that includes: 1) a Watergate spin-echo (Piotto et al., 1992) to refocus the chemical shift evolution after the DIPSI-2 irradiation and to achieve solvent suppression, and 2) an indirect evolution period before the TOCSY pulse-train to record high-resolution spectra in inhomogeneous $B_0$ fields similarly to experiments by Huang et al. (2010) that used the DF combined with spin-echo correlation spectroscopy (SECSY, Nagayama et al. (1979)). In fig. 2b, a 1D spectrum obtained with $t_1 = 0$ in a homogeneous $B_0$ field of 2 mM sucrose and 0.5 mM sodium 3-(trimethylsilyl)propane-1-sulfonate (DSS) in a 90%/10% mixture of $H_2O/D_2O$ (the same sample is used throughout this work). This sample is a standard to test water suppression, with a well dispersed distribution of resonances between 0 ppm and 6 ppm. The inset, that shows part of the spectrum, highlights the small phase distortions at long DIPSI-2 irradiation times ($> 100$ ms), which cannot be simultaneously corrected for all resonances by a linear phase correction. The $^{13}C$ satellites on both sides of the methyl protons (marked with asterisks) correspond to a concentration of 22.5 $\mu$M.

The results of a 2D experiment in an inhomogeneous $B_0$ field are shown in fig. 3a (the line-width at half height measured on the water resonance was about 225 Hz). The coherence selection pathway below the sequence in fig. 2 shows that $B_0$ inhomogeneities that have evolved in $t_1$ should be refocused during the direct dimension at the time $t_2 = t_1$, which leads to the skewed line-shapes. No corrections or special processing protocols were applied to remove phase twists in the spectrum. The absence of those are due to "constructive interference" of neighboring peaks (section 6.5.2 of Ernst et al. (1992)). Moreover,

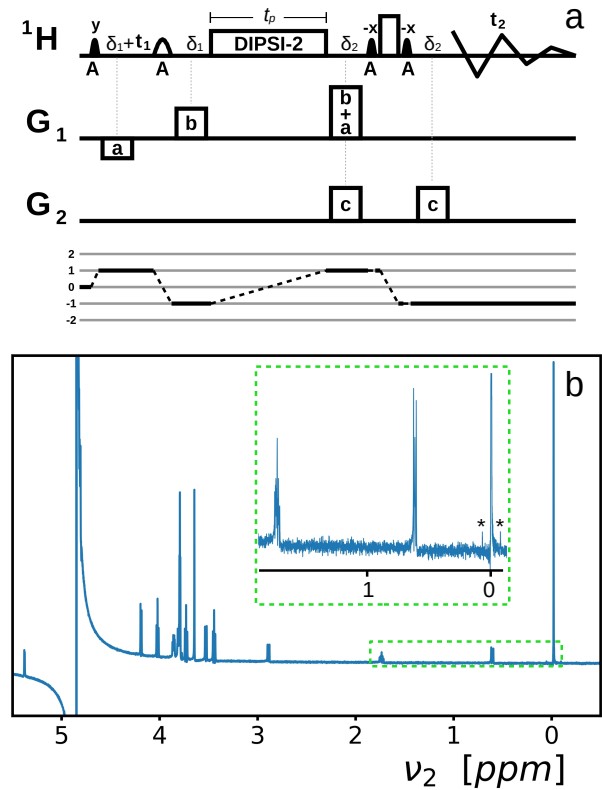

**Figure 2. (a)** Variant of the selective TOCSY pulse sequence that facilitates the transfer of phase coherence by the DF (Pelupessy, 2022b) adapted to record broadband in-phase high resolution spectra in inhomogeneous fields with solvent suppression. Narrow filled and wide open shapes stand for $90°$ and $180°$ rf pulses, respectively. Low amplitude pulses are selectively applied to the abundant $A$ spins, as indicated below the pulses. The DIPSI-2 pulse-train and the rectangular high amplitude pulse are broadband. All pulses are applied along the $x$-axis, unless specified otherwise. $G_1$, $G_2$ and $G_3$, indicate three orthogonal PFG directions, not necessarily $x$,$y$ or $z$. The delays $\delta_i$ accommodate the lengths of the PFGs (in this work, all PFGs had equal length). $t_p$ is the duration of the DIPSI-2 pulse-train, $t_1$ and $t_2$ are the indirect and direct evolution times. The coherence of the $A$ spins is modulated by the two PFGs $\mathbf{G}_a$ and $\mathbf{G}_b$. The amplitude of $\mathbf{G}_a$ must be large enough to quench effects of RD, but as low as possible to diminish losses due to diffusion. The area of $\mathbf{G}_{a+b}$ must be equal to the sum of the areas of $\mathbf{G}_a$ and $\mathbf{G}_b$, the signs of the PFGs are indicated by positive or negative rectangles. In general, a PFG marked $pa + qb$ means a PFG that is equal to the vector addition $p$ times $\mathbf{G}_a$ and $q$ times $\mathbf{G}_b$. Purging PFGs $\mathbf{G}_c$ help to suppress the signal of the $A$ spins. The coherence selection pathway is plotted below the sequence. **(b)** 1D spectrum recorded with $t_1 = 0$ and a DIPSI-2 irradiation time of 200 ms in a homogeneous $B_0$ field. The signals below 3 ppm belong to DSS (0.5 mM), among them at 0 ppm the 9 methyl protons (the $^{13}$C satellites are marked with asterisks). The dispersive peak at 4.85 ppm stems from leftover $H_2O$. All other signals come from sucrose (2 mM). In the supporting information, the assignments are given.

150    even if present, phase twists hardly perturb the results (see supporting information). In the indirect $t_1$ dimension, the spectrum needs to cover the inhomogeneously broadened line-shape (here 1 ppm was used). On the right (fig. 3b), a sliding window

function has been applied in the direct $t_2$ dimension and subsequently the spectrum has been sheared so that the elongated ridges appear perpendicular to the $\nu_2$-axis. The window function $W_i$ consisted of zeroes for all time-points except for a narrow range of points $k = \{-d + o, d + o\}$ where the intensities were multiplied by:

$$W_k = 1 - \sin^{2n}\{\pi(k - bw_2 t_1)/(2d)\}, \tag{6}$$

where $bw_2$ is the bandwidth in the direct dimension (the inverse of the time-increment), and the offset $o$ is the integer nearest to $bw_2 t_1$. The higher the integer value $n$, the closer it is to a rectangular profile. The broader the inhomogeneous line, the sharper the echo and, consequently, the smaller the range $2d$. This function is identical to the amplitude modulation of wide-band, uniform rate and smooth truncation pulses (WURST, Kupce and Freeman (1995)). The variables $d = 300$ and $n = 1$ were optimized empirically (the value of $n$ only slightly affects the result). Application of this window function resembles chunk selection in pure shift NMR (Zangger and Sterk, 1997).

The 1D spectrum in fig. 3c corresponds to the sum of the middle 112 rows (of a total of 512) of the spectrum of fig. 3b. On the side panels, parts of this spectrum (figs. 3d1 and 3e1, corresponding to a very crowded region and a weak and complex multiplet), are compared with the results of a pulse-acquire experiment preceded by saturation of the solvent signal in a homogeneous $B_0$ field (figs. 3d2 and 3e2). Figs. 3d3 and 3e3 show parts of the Fourier transform of the first increment ($t_1 = 0$). The enlargements of figs. 3d1 and 3e1 closely resemble those of figs. 3d2 and 3e2. In the lower spectra, slight distortions due to scalar coupling evolution during the Watergate spin echo are visible and the relative peak intensities do not exactly match those in the center spectra.

The experiment presented in this section is capable of delivering broadband in-phase well-resolved spectra with only small distortions. It shares these characteristics with the experiment of Fugariu et al. (2017), which adds an adiabatic spin-lock to the method of Huang et al. (2010) in order to avoid scalar coupling evolution during the transfer by the DF, but keeps the same effective dipolar Hamiltonian as the one during free precession and achieves the changes of coherence order solely by rf pulses. Since the effective Hamiltonian is scaled during the DIPSI-2 pulse-train, the transfer is slower compared to sequences which rely on prolonged free precession delays. On the other hand, the entire (phase-modulated) magnetization of the solvent contributes to the transfer. If the solute relaxation rates are higher than the ones of the solvent (as is often the case), the new sequence should be advantaged since the losses depend on a mix of longitudinal and transverse relaxation rates. Moreover, contributions from conformational exchange to the relaxation will be attenuated. In vivo, its use may be limited due to the prolonged duration of rf irradiation.

### 3.2 Double quantum transfer

In the original CRAZED sequence (COSY revamped with asymmetric $z$-gradient echo detection) of Warren et al. (1993), intermolecular DQ coherences are converted to SQ ones by a 90° pulse. The DF then converts these multiple-spin SQ coherences to observable one-spin SQ coherences. The appropriate coherence order pathway needs to be selected by a judicious choice of PFGs, in this case a 1:2 ratio of the areas of the PFGs before and after the 90° mixing pulse. This rf pulse is essential, because the dipolar Hamiltonian does not allow for a change in coherence order. As seen in the previous section, this constraint does

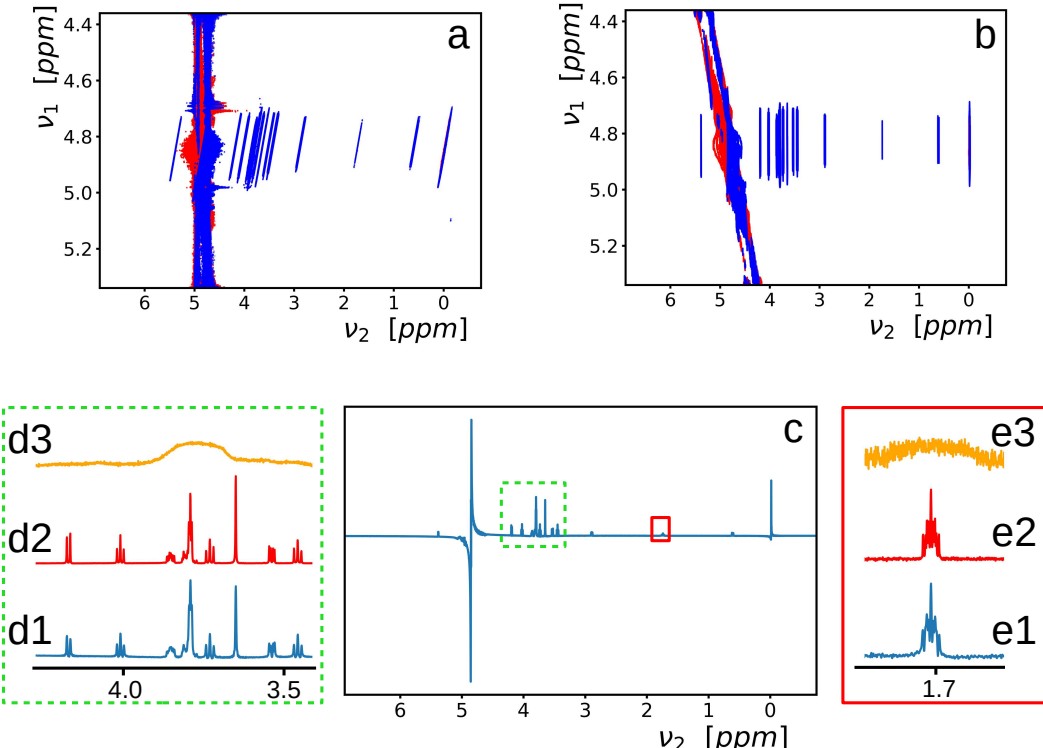

**Figure 3. (a)** 2D spectrum recorded with the pulse sequence of fig. 2a under the same conditions as the 1D spectrum of fig. 2b, except that the $B_0$ field was purposely rendered inhomogeneous (line-widths of about 225 Hz). **(b)** The 2D spectrum of (a) has been tilted and the sliding window function of eq. 6 has been applied. **(c)** The sum of the central 112 rows of (b). **(d1, e1)** Two regions of the spectrum (c) have been enlarged. **(d2, e2)** The corresponding regions of a pulse-acquire spectrum, preceded by saturation of the solvent signal, in a homogeneous $B_0$ field. **(d3, e3)** The same regions from the Fourier transform of the first free induction decay (corresponding to $t_1 = 0$) of the experiment.

185 not apply to the effective Hamiltonian of eq. 5. Hence, it may be interesting to explore the effect of the DF on intermolecular DQ coherences during DIPSI-2 irradiation.

After the first 90° pulse of the sequence in fig. 4a, the magnetization of the $A$ and the $S$ spins is de-phased by a PFG $\mathbf{G}_a$. The commutator of $H_{AS}^{eff}$ of eq. 5 with the lowest order term in the expansion of the density operator, which contains a product of $S$ and $A$ spin operators, results in:

190
$$-i\big[H_{AS}^{eff},(c_aS_x + s_aS_y)(c_aA_x + s_aA_y)\big] = -(3/16)s_{2a}(S_z + A_z)\,, \tag{7}$$

where $c_a$ and $s_a$ stand for $\cos(\tau_a\gamma\mathbf{G}_a\cdot\mathbf{r})$ and $\sin(\tau_a\gamma\mathbf{G}_a\cdot\mathbf{r})$, with $\tau_a$ the effective duration of $\mathbf{G}_a$ and $\mathbf{r}$ the spatial position. In general, for arbitrary numbers $p$ and $q$, $c_{pa+qb}$ stands for $\cos(p\tau_a\gamma\mathbf{G}_a\cdot\mathbf{r} + q\tau_b\gamma\mathbf{G}_b\cdot\mathbf{r})$. Thus, the DIPSI-2 irradiation leads to a transfer of a DQ coherence involving the $A$ and $S$ spins to longitudinal magnetization. A purge gradient followed by a 90° rf

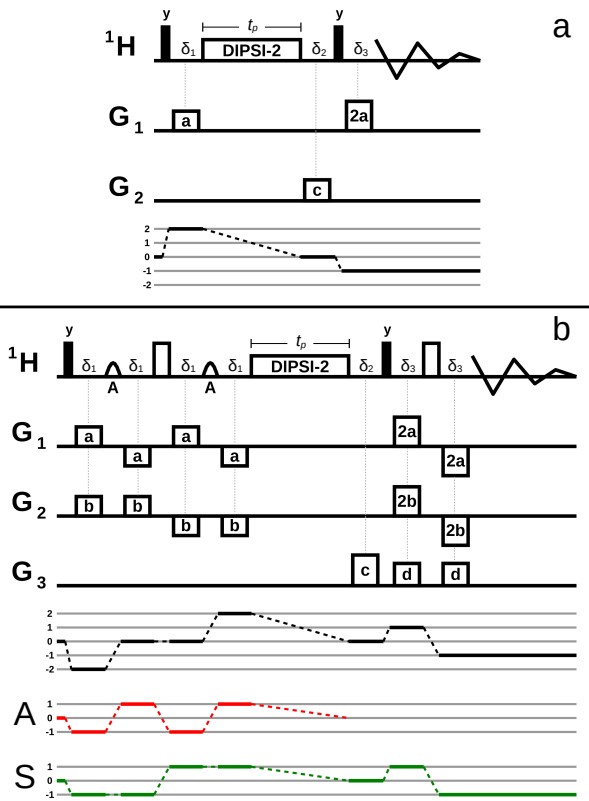

**Figure 4.** Pulse sequences where the DF mediates the transfer from a DQ coherence involving the abundant spins $A$ and the sparse spins $S$ to a longitudinal magnetization during the DIPSI-2 pulse-train. See the caption of fig. 2 for an explanation of the different rf pulses, delays, and PFGs. The coherence selection pathways are plotted below the sequences. **(a)** Non-selective experiment where both the solvent spins $A$ and solute spins $S$ are affected by $\mathbf{G}_a$. After a purging PFG $\mathbf{G}_c$ and a 90° rf pulse, a PFG with twice the area of $\mathbf{G}_a$ is necessary to observe the signal of the $S$ spins. **(b)** Same type of transfer as (a), with separate gradient labeling of spins $A$ by $4\mathbf{G}_a$ and of spins $S$ by $4\mathbf{G}_b$. The signal of the $S$ spins needs to be recovered by the sum of both PFGs. The two PFGs $\mathbf{G}_d$, while not strictly necessary, improve the solvent suppression. On the bottom, the coherence selection pathways of the $A$ (red) and $S$ (green) spins are displayed separately.

pulse and a final PFG $+2\mathbf{G}_a$ or $-2\mathbf{G}_a$ ($p\mathbf{G}_a$ stands for a PFG along the direction of $\mathbf{G}_a$ with an area $p$ times as large), allows one to detect the longitudinal term of the sparse $S$ spins. A phase difference $\phi$ due to chemical shift evolution causes a phase shift $\phi$ in the signal and additional longitudinal components which are not modulated by the gradient.

In fig. 5a, the intensity, $m_{xy} = \sqrt{m_x^2 + m_y^2}$, of the methyl proton resonance of DSS is plotted as a function of the orientation angle $\Theta_{Ga}$ of the PFG $\mathbf{G}_a$ for a DIPSI-2 irradiation time of about 100 ms. The line that goes through the points corresponds to a function $m_{xy}^0(3\cos^2\Theta_{Ga} - 1)/2$, where $m_{xy}^0$ is the intensity recorded with the PFG oriented parallel to $B_0$. The line crosses zero at the so-called magic angle $\Theta = 54.74°$. Although by definition all intensities are positive, for clarity, the intensities of

signals that point in opposite directions when phased identically are plotted with opposite signs. The theoretical curve should match the experimental points only at short irradiation times (i.e., in the linear regime of the buildup).

In fig. 5b, the buildup of intensities is plotted as a function of the irradiation time $t_p$ for orientations parallel ($\Theta_{Ga} = 0°$, blue squares) and perpendicular ($\Theta_{Ga} = 90°$, blue circles) with respect to $B_0$. The blue dot-dashed lines are simulations of these experiments using the approximate solution of the non-linear Bloch eqs. 2 for the evolution of $\mathbf{m}^A$. The intensities for $\Theta_{Ga} = 90°$ at longer irradiation times exceed half of those for $\Theta_{Ga} = 0°$. This is because, when the magnitude of $\omega_d$ is divided by a factor of 2, the transfer is not half as efficient, but rather twice as slow (of course, a slower transfer renders the experiment more sensitive to losses due to relaxation and diffusion). The red crosses are the intensities obtained with the experiment described in Pelupessy (2022b) (i.e., the experiment of fig. 2a, but without refocusing pulses, solvent suppression and an indirect evolution period) for $\Theta_G = 0°$ and a coherence pathway selection $+1 \rightarrow -1$. For clarity, the latter intensities have been divided by 2 since the (simplified) commutator formalism predicts that the initial slopes should differ by a factor 2. This factor applies only for the initial slopes. Neglecting relaxation and diffusion, the transfer reaches a theoretical maximum of $m_{xy}^S = 0.88$ at $t_p \approx 450$ ms for the experiment of fig. 2 and of $m_{xy}^S = 0.32$ at $t_p \approx 320$ ms for the experiments of fig. 4.

In the sequence of fig. 4b, the solvent and solute spins are labeled by different gradients and the chemical shifts are refocused. The $A$ spins are de-phased by $4\mathbf{G}_a$ and the $S$ spins by $4\mathbf{G}_b$. During the DIPSI-2 irradiation, the DF induces the following transfer:

$$-i\Big[H_{AS}^{eff}, (c_{4b}S_x - s_{4b}S_y)(c_{4a}A_x - s_{4a}A_y)\Big] = (3/16)s_{4a+4b}S_z - (1/16)s_{4a-4b}S_z + ... \tag{8}$$

As in the previous sequence, only single spin SQ terms of the $A$ spins suffice for a qualitative description. The first term on the right can be recovered after a 90° pulse by the sum of the two PFGs, $\pm(4\mathbf{G}_a + 4\mathbf{G}_b)$. The solvent senses only $\mathbf{G}_a$ before the DIPSI-2 irradiation so that no additional measures need to be taken to suppress its signal after the DIPSI-2 irradiation and, consequently, the echo before signal detection is shorter than the one in fig. 2. The characteristic frequency $\omega_d$ of the DF depends only on the orientation of $\mathbf{G}_a$.

In fig. 5c, spectra obtained with $\mathbf{G}_a$ along the $z$-axis and $\mathbf{G}_b$ along the $x$-axis for several DIPSI-2 irradiation times are displayed. The buildup curves of three resonances (the methyl resonance of DSS, and two sucrose resonances) are plotted in fig. 5d. The buildup curves are very similar at short times and start to diverge at longer times, probably due to differences in relaxation rates.

### 3.3 Mixed modulations

When applying multiple rf pulses interleaved with PFGs in different directions, the modulation of the magnetization can become rapidly very convoluted. Often, PFGs serve as purging gradients and the parts of the density operator that are de-phased by these PFGs can be discarded. However, magnetization which is modulated in complex patterns can also be implicated in a transfer of phase coherence by the DF, although the conditions for the validity of the theory described in section 2.1 may not strictly apply anymore. In this section, this type of more intricate modulations will be investigated. Several schemes will be

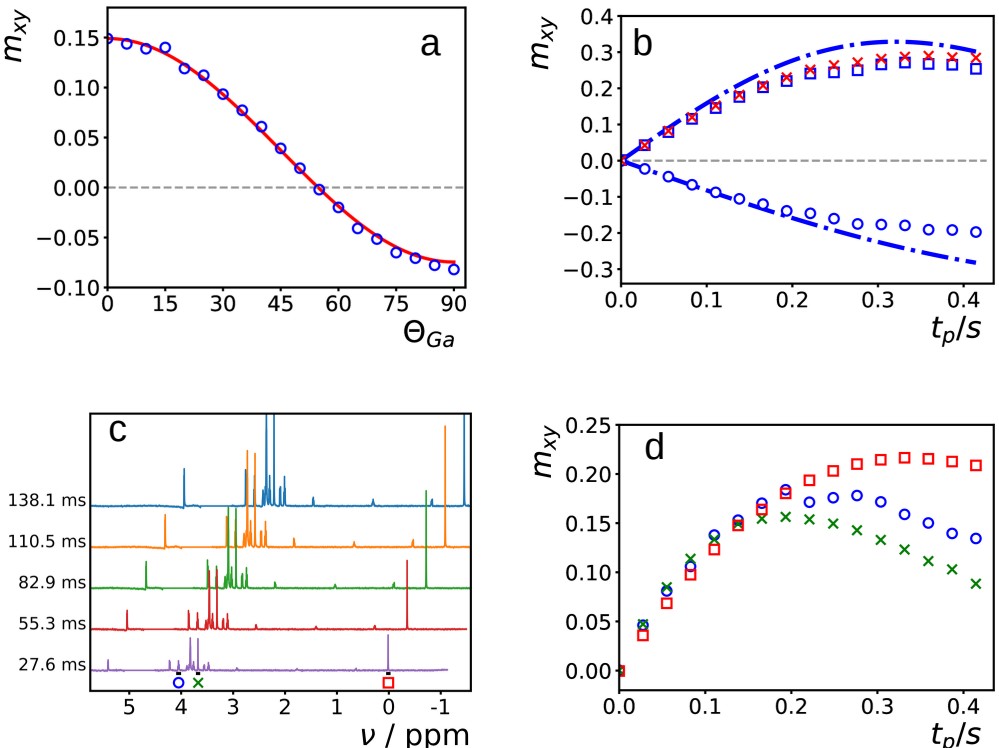

**Figure 5. (a)** Signal intensities of the methyl protons of DSS obtained with the sequence of fig. 4a with a DIPSI-2 irradiation time of 100 ms. The orientation of $\mathbf{G}_a$ was varied while its amplitude was kept constant. While $m_{xy}$ is always positive, it has been multiplied by $-1$ for signals that were opposite with respect to the first one. The red line is a function proportional to $3\cos^2\Theta_{Ga} - 1$ that goes through the first point. **(b)** At the angles $\Theta_{Ga} = 0°$ (blue squares) and $\Theta_{Ga} = 90°$ (blue circles) the evolution of the signal has been recorded as a function of DIPSI-2 irradiation time $t_p$. The dash-dotted lines are simulations as explained in the main text (without taking into account relaxation or translational diffusion). The red crosses are intensities, scaled down by a factor 2, obtained with the experiment described by Pelupessy (2022b). **(c)** Spectra of a solution of 2 mM sucrose and 0.5 mM DSS in $H_2O/D_2O$ (90%/10%) recorded with the experiment of fig. 4b for several irradiation times $t_p$ indicated on the left side of the spectra. The baseline of each spectrum has been corrected separately for the regions on the left and right of the solvent signal which has been digitally removed for clarity. **(d)** The buildup curves of three selected resonances (see supporting information for the assignments) marked at the bottom of (c).

presented where the density operator is prepared in different ways and where the transfer from a $+1$ to a $-1$ coherence order is recorded.

At the start of the sequence of fig. 6a, a PFG $\mathbf{G}_a$ sandwiched between two selective 90° pulses modulates the amplitude of the magnetization of the abundant $A$ spins. It is followed by a purging PFG and another selective pulse and PFG $\mathbf{G}_b$ before DIPSI-2 irradiation. Neglecting the parts of the density operator which are de-phased by $\mathbf{G}_c$, the transfer due to the DF that

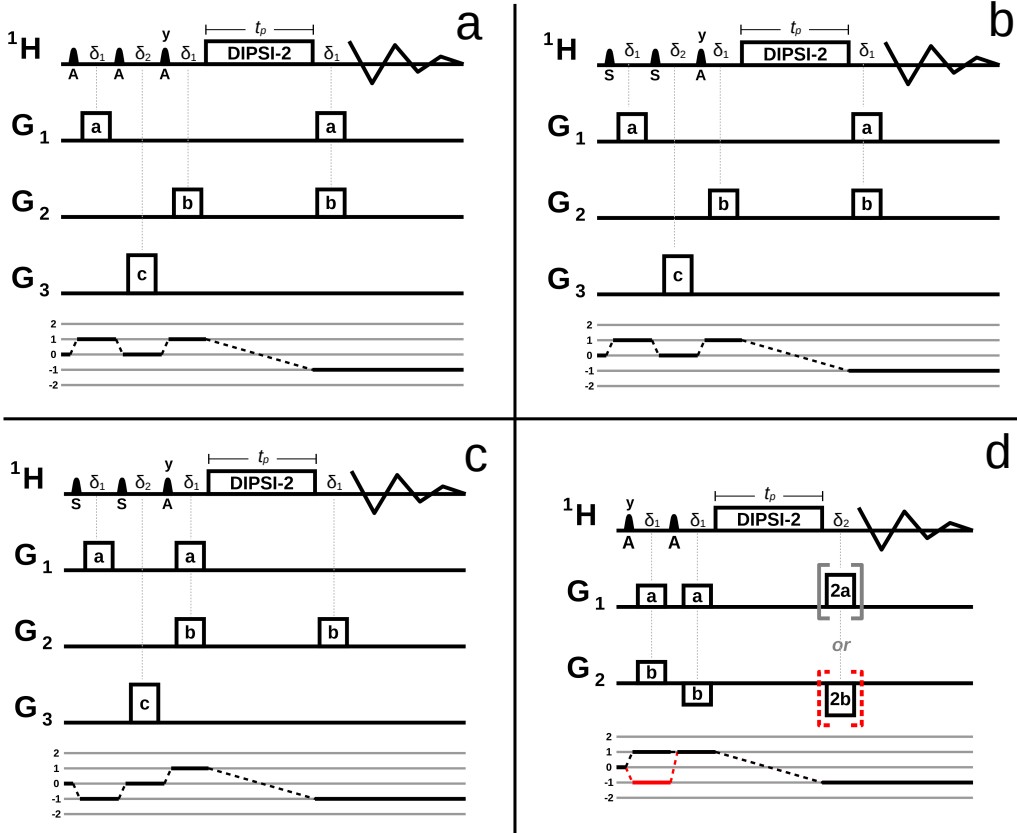

**Figure 6.** Variants of pulse sequences that record the transfer of phase coherence order from $+1$ to $-1$ by the DF. See the caption of fig. 2 for an explanation of the different rf pulses, delays, and PFGs. **(a)** A PFG $\mathbf{G}_a$ sandwiched between two selective $90°$ pulses modulates the amplitude the $A$ magnetization. After a purging PFG $\mathbf{G}_c$ and a $90°$ pulse, $\mathbf{G}_b$ modulates the phase of the $A$ spins. Following the DIPSI-2 irradiation, the newly created coherence on the $S$ spins can be recovered by the two PFGs $\mathbf{G}_a$ and $\mathbf{G}_b$. **(b)** Similar as (a) except that the first two $90°$ pulses are selectively applied on the $S$ spins. **(c)** Similar to (b), except that the gradient $\mathbf{G}_a$ has been moved in front of the DIPSI-2 irradiation. The coherence selection pathway is different compared to (a) and (b). **(d)** The combination of PFGs and rf pulses before the DIPSI-2 pulse-train create two DFs, due to a modulation of the magnetization of $A$ in different directions, both of them simultaneously mediating a transfer of phase coherence. Either the modulation due to $\mathbf{G}_a$ or the one due to $\mathbf{G}_b$ is recovered. When the gradient $2\mathbf{G}_a$ is used, the selected coherence order after the first rf pulse is $+1$, while for $-2\mathbf{G}_b$ it is $-1$ (red line).

occurs during the DIPSI-2 irradiation can be described as follows:

$$-i\Big[H_{AS}^{eff}, S_z(c_a c_b A_x + c_a s_b A_y)\Big] = (2/16)c_a s_b S_x + (4/16)c_a c_b S_y + ... \tag{9}$$

Using the relations $c_a c_b = (c_{b+a} + c_{b-a})/2$ and $c_a s_b = (s_{b+a} + s_{b-a})/2$, the DF can be decomposed into two fields, one originating from the half of the magnetization of the $A$ spins that is de-phased by $\mathbf{G}_b - \mathbf{G}_a$ and the other from the half that is de-phased by $\mathbf{G}_b + \mathbf{G}_a$. If the gradients $\mathbf{G}_b$ and $\mathbf{G}_a$ are oriented either parallel or perpendicular to $B_0$, both fields are

characterized by the same value of $\omega_d$. With the first two pulses applied selectively to the $S$ instead of to the $A$ spins, shown in fig. 6b, the operator analysis remains the same so that eq. 9 also describes the transfer in this case.

The experiments of figs. 6a and 6b were performed with $\mathbf{G}_b$ along $z$ and $\mathbf{G}_a$ along $x$ and a DIPSI-2 irradiation time of 100 ms. The areas of the PFGs have been varied to change the angle $\Theta_{Ga+Gb}$ of the vector addition of $\mathbf{G}_a$ and $\mathbf{G}_b$ with respect to $B_0$ while keeping the amplitude constant. The transfer efficiency is plotted as a function of $\Theta_{Ga+Gb}$ in figs. 7a and 7b for the experiments of figs. 6a and 6b, respectively. Only in the first experiment where all selective pulses are applied to $H_2O$ the typical $(3\cos^2\Theta - 1)$ dependence is observed. The direction of the spatial modulation of the $A$ spins is important, which does not

change with $\Theta_{Ga+Gb}$ in the experiment of fig. 6b since the first two pulses are applied on the $S$ spins. While for $intra$molecular two-spin operators it does not matter how the amplitude- or phase-modulation has been created, for $inter$molecular two-spin operators eq. 9 does not provide the full picture. For a spin $S$ at a given position, one has to consider the dipolar interactions with all spins $A$ (Lee et al., 1996), whose spatial modulation is different for the two experiments. The $(3\cos^2\Theta_{Ga+Gb} - 1)$ dependence which is absent in the experiment of fig. 6b is restored in the sequence of fig. 6c, where the second gradient $\mathbf{G}_a$ is

put before the DIPSI-2 block instead of behind, as shown in fig. 7c.

For the experiment of fig. 6a, fig. 7d shows the transfer as a function of the DIPSI-2 irradiation time at angles $\Theta_{Ga+Gb}$ = 15° (blue circles) and $\Theta_{Ga+Gb}$ = 80° (green triangles). The dashed dotted curves are simulations assuming $\omega_d$ is constant throughout the irradiation time, which may not be entirely correct since the spatial pattern of the modulations of the magnetization of $A$ will change over time (although the modulations will remain in the plane spanned by the vectors in the directions

of $\mathbf{G}_b - \mathbf{G}_a$ and $\mathbf{G}_b + \mathbf{G}_a$). The red crosses correspond to the experiment of fig. 6c with $\Theta_{Ga+Gb}$ = 15°. The buildup is almost indistinguishable from the one in blue circles. The simulated curves for the latter experiment (here $\omega_d$ is constant since the magnetization of $A$ is modulated in one direction only) are almost identical to the previous ones, indicating that the assumption of a constant $\omega_d$ used for the simulations of experiments of fig. 6a is a reasonable approximation for these DIPSI-2 irradiation times.

In the experiments above, the modulation of the magnetization was decomposed into two directions, not necessarily orthogonal, each at the same angle with respect to $B_0$. This is different in the following experiment. Just before the DIPSI-2 irradiation in the pulse sequence of fig. 6d, the two-spin $S_z A_z$ term of $\rho_{eq}$ in eq. 4, has evolved into :

$$\rho = S_z\{A_x c_{a+b} c_{a-b} + A_y c_{a+b} s_{a-b}\} = S_z\{A_x(c_{2a} + c_{2b})/2 + A_y(s_{2a} - s_{2b})/2\} \ . \tag{10}$$

A term containing $S_z A_z$ (created by the second selective pulse) has been left out since it commutes with the effective dipolar

Hamiltonian of eq. 5 and will not lead to an observable signal. Eq. 10 corresponds to a situation where half of the magnetization is modulated in the direction of $\mathbf{G}_a$ the other half in the direction of $\mathbf{G}_b$. Experiments were performed with $\mathbf{G}_a$ parallel and $\mathbf{G}_b$ perpendicular to $B_0$, with different irradiation times. The blue circles in fig. 8 correspond to the signal intensities of the methyl protons of DSS while using a PFG $2\mathbf{G}_a$ before detection. The green squares correspond to intensities recorded using a PFG $-2\mathbf{G}_b$. The signal intensities represented by the red crosses were recorded under the same circumstances as the second

set of experiments (i.e., with $-2\mathbf{G}_b$ before acquisition), except that $\mathbf{G}_a$ was perpendicular to $both$ $\mathbf{G}_b$ $and$ $B_0$. The intensities of the latter experiments at longer times are slightly higher than the intensities of the green squares. This is likely caused by

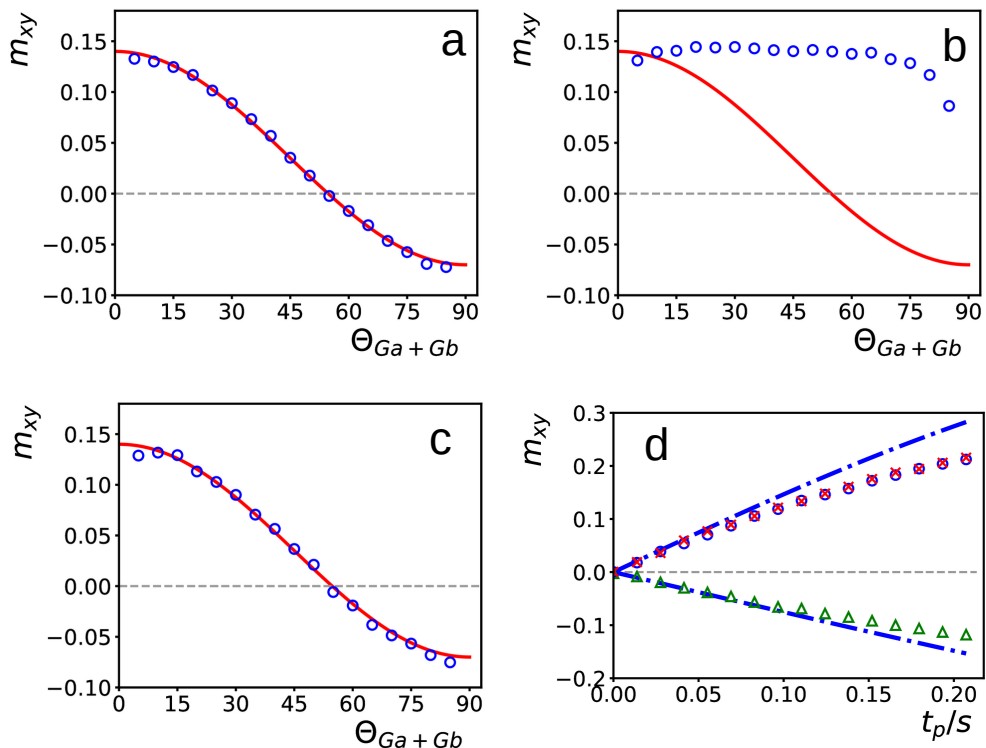

**Figure 7.** **(a-c)** Signal intensities of the methyl protons of DSS obtained with the pulse sequences of fig. 6a-c, respectively. A DIPSI-2 irradiation time of about 100 ms has been used. The angle $\Theta_{Ga+Gb}$, i.e. the angle between the vector addition of PFGs $\mathbf{G}_a$ and $\mathbf{G}_b$ and $B_0$ has been varied between $5°$ and $85°$, while keeping the amplitude constant. The red lines, which correspond to a function $0.14 \times (3\cos^2\Theta_{Ga+Gb} - 1)/2$, match the experimental points in (a) and (c). **(d)** Signal intensities as a function of DIPSI-2 irradiation time $t_p$, recorded under the same conditions as in (a) with $\Theta_{Ga+Gb} = 15°$ (blue circles) and $80°$ (green triangles) and as in (c) with $\Theta_{Ga+Gb} = 15°$ (red crosses). The dash dotted curves are simulations as explained in the main text.

the larger perturbation of the longitudinal magnetization of the $S$ nuclei that occurs when one of the modulations is parallel to the $B_0$ field.

When the magnetization is modulated in only one dimension the strength of the DF scales with the local value of $\mathbf{m}^A$. This
is not true anymore with the last scheme of this section, since the DF can be decomposed into two different parts each with its own characteristic value of $\omega_d$. Remarkably, for a given position in the sample the magnetization of the solute $\mathbf{m}^S$ may be affected by the DF albeit the solvent magnetization vanishes ($\mathbf{m}^A = 0$) at that position. Another counterintuitive feature is that even if the vector addition of the PFGs after the first rf pulse *and* the vector addition of the PFGs after the second rf pulse both point along the magic angle with respect to $B_0$, it is still possible to observe a transfer of phase coherence. Hence, application
of PFGs oriented along the magic angle do not always suffice to suppress effects of the DF.

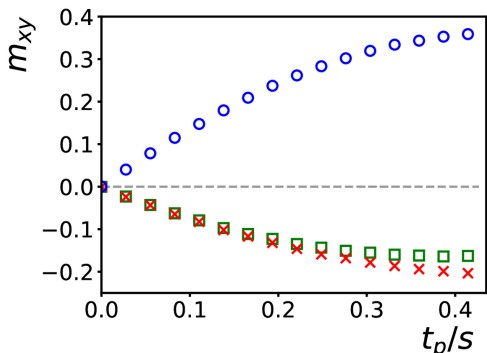

**Figure 8.** Buildup of the intensities of the methyl protons of DSS obtained with the experiment of fig. 6d at different DIPSI-2 irradiation times $t_p$. $\mathbf{G}_a$ was parallel to $B_0$, $\mathbf{G}_b$ perpendicular. The blue circles were recorded with $2\mathbf{G}_a$ as the last gradient before acquisition, the green squares with $-2\mathbf{G}_b$. The red crosses have been recorded under the same conditions as the green squares, except that orientation of $\mathbf{G}_a$ was perpendicular to $B_0$ and $\mathbf{G}_b$.

## 4   Experimental parameters

All experiments were acquired in a $B_0$ field of 18.8 T (800 MHz proton frequency) at 290 K, with a probe equipped with coils to generate PFGs along three orthogonal axes. The rf amplitude $\omega_1/2\pi$ during the DIPSI-2 pulse-train was 8.33 kHz. The selective pulses on either the solvent or the methyl protons of DSS had Gaussian shapes and a length of 5 ms (for both the 90° and 180° pulses), except the two 90° pulses in the Watergate scheme of fig. 2 which had a sinc profile and a duration of 2 ms. The shaped pulses that were applied on the equilibrium magnetization of the abundant spins $A$ were calibrated separately to compensate for effects of RD. All PFGs had smoothed square profiles and durations of 1 ms. $G_1$, $G_2$ and $G_3$ indicate orthogonal gradient channels. The amplitudes of the PFGs were $G_a = 1.6$ G/cm, $G_b = 6.2$ G/cm, $G_{a+b} = 7.8$ G/cm and $G_c = 32$ G/cm for the experiments of figs. 2 and 3, $G_a = 7.8$ G/cm, $G_c = 27$ G/cm for figs. 5a and 5b, $G_a = 1.95$ G/cm, $G_b = 1.95$ G/cm, $G_c = 27$ G/cm and $G_d = 10$ G/cm for figs. 5c and 5d, $\|\mathbf{G}_a + \mathbf{G}_b\| = 7.8$ G/cm and $G_c = 32$ G/cm for fig. 7, $G_a = 3.9$ G/cm and $G_b = 3.9$ G/cm for fig. 8. For all experiments, 8192 complex points were acquired with a bandwidth of 12 ppm. All signal intensities, that have been quantified, have been normalized either to the spectrum of the methyl protons after a selective excitation or to a broadband pulse-acquire experiment preceded by saturation of the solvent signal.

The 2D spectra of fig. 3 has been acquired with 256 $t_1$ increments, a bandwidth of 1 ppm and 4 scans per increment in the indirect dimension and a repetition time of 11 s (in the supporting information, a similar experiment recorded with only 1 scan, a repetition time of 3 s, and a DIPSI-2 irradiation time of 100 ms is shown). For these spectra, the experimental data have been doubled in each dimension by zero padding. The spectrum of fig. 3a has been obtained by a 2D Fourier transform without any apodization. For the spectrum of fig. 3b, the same data have been multiplied by the window function of eq. 6 and subsequently Fourier transformed along the indirect $t_1$ dimension. A first order phase correction along the direct $t_2$ dimension, proportional to the $\nu_1$ position, was then applied to shear the spectrum, followed by a Fourier transform along the same dimension.

## 5   Conclusions

In this work, we have investigated several aspects of the transfer of phase coherence by the dipolar field during rf irradiation sequences that have been developed for total correlation spectroscopy, in particular the DIPSI-2 pulse-train. Theoretical expressions for the evolution of the solvent spins under continuous rf irradiation have been derived, which permits efficient simulations of the transfer process. A remarkable feature is that, under these conditions, the DF can cause not only a transfer, but also a change of coherence order. Nevertheless, the formalism developed by Warren and coworkers (Lee et al., 1996) can still be used – taking into account the effective Hamiltonian of eq. 5 – to describe and design pulse sequences. An experiment for the acquisition of broadband in-phase high-resolution spectra in inhomogeneous fields has been presented. In this experiment, the transfer takes place from a SQ coherence of the abundant solvent spins to another SQ coherence of the sparse solute spins. Additionally, alternative coherence order pathways have been investigated: a DQ coherence involving both solvent and solute spins can be converted by the DF into longitudinal magnetization. Two pulse sequences to record this transfer have been introduced. In the first sequence, the DQ coherence is de-phased by a PFG, while the SQ coherence, which is detected, is re-phased with a PFG twice as strong. In the second sequence, the solvent and solute parts of the DQ coherence are de-phased by two different PFGs. The SQ coherence is now re-phased by the sum of these two PFGs. The latter sequence has the advantage of resulting in broad-band in-phase spectra, without the need for additional water-suppression before acquisition. In the last part, more complex modulation patterns of the magnetization have been investigated. The importance of how the magnetization of the different spins has been modulated in *inter*molecular multiple-spin operators has been explored with three closely resembling pulse sequences. Finally, by combining several pulsed field gradients and rf pulses the DF has been tailored in such a manner that it can be decomposed into different components that have their own spatial modulation and, hence, can simultaneously bring about different transfer processes.

In the different sequences, DIPSI-2 pulse-trains have been applied. Other TOCSY mixing sequence can be used, although one has to take into account the different effective dipolar Hamiltonians that characterize these sequences (Kramer and Glaser, 2002). These effects could be suppressed by the use of tailored mixing sequences (Klages et al., 2007), or the use of PFGs along the magic angle. However, the results in section 3.3 caution against the latter option for sequences with many pulses and PFGs. The effects of the transfer by the scalar couplings during the TOCSY sequence have not been taken into account. This is reasonable because either an uncoupled nucleus has been probed or because the solute coherences had all the same phase or were all along the $z$-axis before the DIPSI-2 irradiation, which minimizes the effects of the transfer by scalar couplings (Braunschweiler and Ernst, 1983). In sequences where the chemical shifts of the solute spins evolve before the TOCSY pulse-train, a more complex behavior is expected.

## Appendix A: Evolution of the magnetization under continuous on-resonance irradiation

Consider the case of a constant on-resonance rf field. Without loss of generality, the rf field can be oriented along the $x$-axis. Hence, for the abundant $A$ spins, the set of equations 2 can be written as:

$$\dot{m}_x^A = -2\alpha m_z^A m_y^A,$$
$$\dot{m}_y^A = (2\alpha m_x^A - \omega_{1x}) m_z^A, \tag{A1}$$
$$\dot{m}_z^A = \omega_{1x} m_y^A,$$

where $\alpha = 3\omega_d/4$. In terms of:

$$m_+^A = m_y^A + i m_z^A,$$
$$m_-^A = m_y^A - i m_z^A, \tag{A2}$$

eqs. A1 become:

$$\dot{m}_x^A = i\alpha(m_+^A m_+^A - m_-^A m_-^A)/2,$$
$$\dot{m}_+^A = i\alpha m_x^A m_-^A + i(\omega_{1x} - \alpha m_x^A)m_+^A, \tag{A3}$$
$$\dot{m}_-^A = -i\alpha m_x^A m_+^A - i(\omega_{1x} - \alpha m_x^A)m_-^A,$$

Usually, $m_+$ and $m_-$ are defined in terms of transverse operators. This traditional definition could have been kept by transforming to a tilted frame. Switching to a rotating frame around the $x$-axis:

$$m_+'^A = e^{-i\omega_{1x}t} m_+^A \Longrightarrow m_+^A = e^{i\omega_{1x}t} m_+'^A,$$
$$m_-'^A = e^{i\omega_{1x}t} m_-^A \Longrightarrow m_-^A = e^{-i\omega_{1x}t} m_-'^A, \tag{A4}$$

one obtains:

$$\dot{m}_x^A = i\alpha(e^{2i\omega_{1x}t} m_+'^A m_+'^A - e^{-2i\omega_{1x}t} m_-'^A m_-'^A)/2,$$
$$\dot{m}_+'^A = i\alpha e^{-2i\omega_{1x}t} m_x^A m_-'^A - i\alpha m_x^A m_+'^A, \tag{A5}$$
$$\dot{m}_-'^A = -i\alpha e^{2i\omega_{1x}t} m_x^A m_+'^A + i\alpha m_x^A m_-'^A.$$

When the oscillating components can be neglected (i.e., when $|\omega_{1x}| \gg |\alpha|$), the solution is a nutation around the $x$-axis with an angular frequency of $\alpha m_x^A(0)$ which in the original frame results in:

$$m_x^A = m_x(0)$$
$$m_y^A = \cos\left\{\omega_{1x}t - \alpha m_x^A(0)t\right\} m_y^A(0) - \sin\left\{\omega_{1x}t - \alpha m_x^A(0)t\right\} m_z^A(0), \tag{A6}$$
$$m_z^A = \cos\left\{\omega_{1x}t - \alpha m_x^A(0)t\right\} m_z^A(0) + \sin\left\{\omega_{1x}t - \alpha m_x^A(0)t\right\} m_y^A(0).$$

*Code and data availability.* The python pulse program used for different simulations can be found in the supporting information.

*Author contributions.*

*Competing interests.* The author has no conflicts of interest to declare.

*Acknowledgements.* I thank Geoffrey Bodenhausen for careful reading and correcting the manuscript and Kirill Sheberstov for fruitful
discussions.

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
