# Peer review of "Various facets of intermolecular transfer of phase coherence by nuclear dipolar fields"

_Magnetic Resonance, 2023_

## Referee Comment (RC3)

[referee-annotated manuscript omitted]

---

## Community Comment (CC1)

$$\underset{\sim}{H} = \int d^3\underset{\sim}{r}' \; \underset{\sim}{\nabla} \left[ \underset{\sim}{M}(\underset{\sim}{r}') \cdot \underset{\sim}{\nabla} \frac{1}{|\underset{\sim}{r}-\underset{\sim}{r}'|} \right] \qquad = \underset{\sim}{\nabla} \Phi$$

$$\frac{1}{|\underset{\sim}{r}-\underset{\sim}{r}'|} = \frac{1}{2\pi^2} \int d^3\underset{\sim}{k} \; \frac{1}{k^2} \, e^{i\underset{\sim}{k}\cdot(\underset{\sim}{r}-\underset{\sim}{r}')}$$

$$\Phi = \frac{1}{2\pi^2} \int d^3\underset{\sim}{k} \; \frac{i\,\widehat{\underset{\sim}{M}}(\underset{\sim}{k})\cdot\underset{\sim}{k}}{k^2} \, e^{i\underset{\sim}{k}\cdot\underset{\sim}{r}}$$

$$\underset{\sim}{H} = \frac{1}{(2\pi)^3} \int d^3\underset{\sim}{k} \left\{ - 4\pi \, \frac{\underset{\sim}{k}(\underset{\sim}{k}\cdot\widehat{\underset{\sim}{M}}(\underset{\sim}{k}))}{k^2} \right\} e^{i\underset{\sim}{k}\cdot\underset{\sim}{r}}$$

$$= -\frac{4\pi}{3} \underset{\sim}{M}(\underset{\sim}{r}) + \int_{\underset{\sim}{r}\neq\underset{\sim}{r}'} d^3\underset{\sim}{r}' \; \frac{3[\underset{\sim}{M}(\underset{\sim}{r}')\cdot(\underset{\sim}{r}-\underset{\sim}{r}')](\underset{\sim}{r}-\underset{\sim}{r}') - \underset{\sim}{M}(\underset{\sim}{r}')}{|\underset{\sim}{r}-\underset{\sim}{r}'|^5 \qquad |\underset{\sim}{r}-\underset{\sim}{r}'|^3}$$

$$\underset{\sim}{B} = - \underset{\sim}{\nabla} \times \int d^3\underset{\sim}{r}' \; \underset{\sim}{M}(\underset{\sim}{r}') \times \underset{\sim}{\nabla} \frac{1}{|\underset{\sim}{r}-\underset{\sim}{r}'|}$$

$$= \frac{1}{(2\pi)^3} \int d^3\underset{\sim}{r}' \; 4\pi \, \frac{\underset{\sim}{k}\times(\widehat{\underset{\sim}{M}}\times\underset{\sim}{k})}{k^2} \, e^{i\underset{\sim}{k}\cdot\underset{\sim}{r}}$$

$$= \frac{1}{(2\pi)^3} \int d^3\underset{\sim}{r}' \; 4\pi \left\{ \underset{\sim}{M} - \frac{\underset{\sim}{k}(\underset{\sim}{k}\cdot\widehat{\underset{\sim}{M}}(\underset{\sim}{k}))}{k^2} \right\} e^{i\underset{\sim}{k}\cdot\underset{\sim}{r}}$$

$$\underset{\sim}{H} + \frac{4\pi}{3}\underset{\sim}{M} = B - \frac{8\pi}{3}M = \int_{\underset{\sim}{r}\neq\underset{\sim}{r}'} d^3\underset{\sim}{r}' \left[ \text{dipole} \right]$$

- Dipole at origin

$$\frac{3(\underset{\sim}{M}\cdot\underset{\sim}{r})\underset{\sim}{r} - \underset{\sim}{M}}{r^5} \quad \frac{\underset{\sim}{M}}{r^3} = \frac{1}{r^3}\left(3(\underset{\sim}{M}\cdot\hat{\underset{\sim}{r}})\hat{\underset{\sim}{r}} - \underset{\sim}{M}\right)$$

M fixed   and   average over $\varphi$   with   $\hat{\underset{\sim}{r}}(\varphi) = R_z(\varphi)\hat{\underset{\sim}{r}}(0)$

- $\hat{\underset{\sim}{r}}(\varphi) = (\underset{\sim}{n}\cdot\hat{\underset{\sim}{r}})\underset{\sim}{n} + c\varphi(\hat{\underset{\sim}{r}} - \underset{\sim}{n}(\underset{\sim}{n}\cdot\underset{\sim}{r}))$

$$- s\varphi(\underset{\sim}{n}\times\underset{\sim}{r})$$

- after LOTS of vector algebra          for $\underset{\sim}{n} = \underset{\sim}{e}_z$ !

$$\langle 3(\underset{\sim}{M}\cdot\hat{\underset{\sim}{r}})\hat{\underset{\sim}{r}} - \underset{\sim}{M}\rangle = \frac{\frac{1}{2}(3\hat{r}_z^2 - 1)\{3M_z\underset{\sim}{e}_z - \underset{\sim}{M}\}}{r^3}$$

- an easier method

$$E = \frac{1}{r^3}\{3(\underset{\sim}{M}\cdot\hat{\underset{\sim}{r}})(\underset{\sim}{N}\cdot\hat{\underset{\sim}{r}}) - \underset{\sim}{M}\cdot\underset{\sim}{N}\}$$

$$= \frac{1}{r^3}\{M_i(3\hat{r}_i\hat{r}_j - \delta_{ij})N_j\}$$

tensor notation

$$= \frac{1}{r^3}M_i D_{ij} N_j$$

- The energy is an invariant, but we can break the symmetry and transform only $D_{ij}$

$$\langle R_z(\varphi) D_{ij} R_z(-\varphi)\rangle = \begin{pmatrix} \frac{3}{2}(\hat{r}_x^2 + \hat{r}_y^2) - 1 & 0 & 0 \\ 0 & \frac{3}{2}(\hat{r}_x^2 + \hat{r}_y^2) - 1 & 0 \\ 0 & 0 & 3\hat{r}_z^2 - 1 \end{pmatrix}$$

- Again, this produces

$$\frac{1}{2}\left(3\hat{r}_z^2 - 1\right)\left\{3M_z N_z - (\underset{\sim}{M}\cdot\underset{\sim}{N})\right\}$$

for a general averaging about an axis $\underset{\sim}{n}$

$$\frac{1}{2}\left(3(\underset{\sim}{\hat{r}}\cdot\underset{\sim}{n})^2 - 1\right)\left\{3(\underset{\sim}{M}\cdot\underset{\sim}{n})\underset{\sim}{n} - \underset{\sim}{M}\right\} \quad ✡$$

- NB   we could also start with

$$\hat{r}_i\left(3M_i N_k - (\underset{\sim}{M}\cdot\underset{\sim}{N})\delta_{ik}\right)r_k$$

and apply the same trick.

- We can now apply this to the Fourier transform pair

$$\frac{1}{(2\pi)^3}\int d\underset{\sim}{k}\; e^{i\underset{\sim}{k}\cdot\underset{\sim}{r}}\; 4\pi\left\{\frac{(\underset{\sim}{\hat{k}}\cdot\hat{\underset{\sim}{M}}(\underset{\sim}{k}))\underset{\sim}{\hat{k}}}{\underset{\sim}{\hat{k}}\cdot\underset{\sim}{\hat{k}}}\right\} = \frac{4\pi}{3}\underset{\sim}{M}(\underset{\sim}{r})$$

$$+ \int d^3\underset{\sim}{r}'\left\{\frac{\underset{\sim}{M}(\underset{\sim}{r}')}{|\underset{\sim}{r}-\underset{\sim}{r}'|^3} - \frac{3(\underset{\sim}{r}-\underset{\sim}{r}')[\underset{\sim}{M}(\underset{\sim}{r}')\cdot(\underset{\sim}{r}-\underset{\sim}{r}')]}{|\underset{\sim}{r}-\underset{\sim}{r}'|^5}\right\}$$

to obtain

$$\frac{1}{(2\pi)^3}\int d\underset{\sim}{k}\; e^{i\underset{\sim}{k}\cdot\underset{\sim}{r}}\; \frac{4\pi}{3}\frac{1}{2}\left(3\frac{k_z^2}{k^2} - 1\right)\left(3\hat{M}_z(\underset{\sim}{k})\underset{\sim}{e}_z - \hat{\underset{\sim}{M}}(\underset{\sim}{k})\right)$$

$$= \int d^3\underset{\sim}{r}'\; \frac{-\frac{1}{2}(3r_z^2 - 1)}{|\underset{\sim}{r}-\underset{\sim}{r}'|^3}\left\{3M_z(\underset{\sim}{r}')\underset{\sim}{e}_z - \underset{\sim}{M}(\underset{\sim}{r}')\right\}$$

✡ This trick can also be used for the expression in $k$ space

---

## Author Comment (AC1)

Dear Dr. Gan,

I thank the reviewers and Tom Barbara for their insightful comments. I have addressed their points in an extensively revised manuscript. Please find below my answers to their comments. For clarity, I have numbered Malcolm Levitt's and Norbert Mueller's comments. The answers and details of the changes in the manuscript are in blue below each comment. I hope this version is suitable for publication in Ampere's Magnetic Resonance.
Kind regards,

Philippe Pelupessy

**Malcolm Levitt**

**1.** the wording is sometimes rather obscure. For example, in the abstract, we read: "the dipolar field can be decomposed into two components, each at the helm of its own transfer". This sounds rather fine, but I have really no idea what it means. Can this be reworded in a more meaningful way, even if less euphonious?
*The wording has been changed, hopefully it is clearer now.*

**2.** Equations (1) and (2) and the neighbouring text underpin this work, but they are discussed very briefly indeed. Take, for example "a PFG oriented at an angle.." - the spatial variation and field direction should be specified more carefully and rigorously. It is well-known that the distant dipolar field is non-local in nature, so that the field at a certain point depends on the magnetization of points which can be anywhere in the sample and distant from the point of interest. So the equations which are used here, which are stated to be "local", rely on some very particular circumstances and approximations. This should be made explicit.
*I have specified the direction and the assumptions more rigorously in the first few paragraphs of the theory section.*

**3.** On page 4 it is stated that only two-spin operators are needed. Why is this?
*This is due to that there are only single quantum terms of the A spins involved in the transfer (and because only a qualitative analysis is performed). A sentence has been added in the manuscript to clarify this point.*

**4.** The figures are quite poor. The spectra are too small and crowded, the spectra are unassigned, and often have no scale. With very careful reading of the caption and text, I could figure out what is what, but the reader should really not have to go to such trouble. The worst one is probably figure 3. It took me about 6 readings to figure out what all three spectra in pane (e) are. I challenge a reader to do it in less, starting cold. I recommend that the figures are made much larger, and the captions made much clearer, explaining each entry in linear sequence, with all individual spectra clearly labelled, and explained in the caption, with no ambiguity and digression - on the lines of (a, top spectrum): xxx; (a, middle spectrum): yyy, etc. etc.
*The spectrum in figure 2 has been enlarged and several figures have been reformatted from single column to double column. Scales have been added to the insets. All individual spectra in figure 3 are now labeled. In the caption of figure 2, I describe the 1D spectrum in more detail (see also the*

*answers to Norbert Mueller's comments for other improvements). And a 1D spectrum with the assignments has been added to the supporting information.*

**5.** The difficult last section of the paper describing the variations in fig.6 is undoubtedly interesting but I struggled to see the point of it, except to display the impressive agreement between experiment and calculation in a complicated case. Maybe the author can make the motivation more obvious. If not, I think one should consider dropping this section.
*This section has been changed quite a bit (see also the answers to Norbert Mueller's comments). I hope the motivation is more comprehensive now.*

**6.** There are a few minor typos such as "undistinguishable" which should be "indistinguishable" (for no particular reason, but that's the way it is).
*I have corrected this.*

**Warren Warren**

**1.** The classical manifestation of these effects has more commonly been referred to as the distant dipolar field (DDF) as it is the portion of the solution dipolar couplings that is not eliminated by diffusion on an NMR timescale. That notation, agreed upon by the groups working in this field about two decades ago, is a compromise that does homage to the still-older but more confusing "dipolar demagnetizing field" in the earliest papers. I would encourage that language instead of "dipolar field" which is baffling to people who are not familiar with these problems.
*In the choice of "dipolar" rather than "distant dipolar", I followed the terminology used by Jean Jeener. I prefer to keep it this way. "Distant" is not very well defined, and using the same acronym as the older one (which may still be useful in some circumstances) is in my opinion confusing. Nevertheless, I should have mentioned both "dipolar demagnetizing field" and "distant dipolar field" in the introduction, which has now been done.*

**2.** It is extremely difficult to tell what in this paper is new. As far as I can tell, the novelty comes from using sequences that are a bit different than what has been explored, including prior work by the author. For example, coherence transfer had been shown in many other contexts, as has compensation for inhomogeneous broadening. Much of what is here is simply a rehash of mathematical formalism that is now decades old, and while I don't think that needs to be deleted, a much clearer delineation is needed. The abstract and the introduction need to be much more focused on separating out what is new.
*The abstract has been rewritten. In the introduction there is now a clear separation. I also think that in the theory section new and old work are now better distinguishable by the reader.*

**3.** The figures in general are not of sufficient quality or clarity.
*The quality of the figures has improved. See also the responses to the other reviewers.*

**Norbert Mueller**

**1.** The equations appear flawless and the quoted references are highly adequate. Some definitions/assumptions, most prominently though maybe trivially, that the context is spin ½ systems only, need to be stated more clearly.
*I have added a statement that the theory is valid for spin ½*

**2.** The text and presentation would benefit from improvements with respect to clarity and maybe a more concise title would be in order. The appendix might be moved to the supplementary information.
*I prefer to keep the appendix in the main manuscript. I think the clarity has improved, see also the answer to the other comments.*

**3.** Overall the structure of the manuscript wrt. to the purpose and motivation of the new experiments introduced should be improved. In the current state it is a relatively lose concatenation of experiments related by their reliance on dipolar field effects. There should also be some justification for the choice of the sample.
*I tried to improve the overall structure of the manuscript and the motivation (see answer to the other comments). Nevertheless, the description as "a relatively lose concatenation of experiments related by their reliance on dipolar field effects" remains appropriate. Some justification of the choice of sample has been added.*

**4.** The Conclusion Section is a bit terse. The benefits and insights provided by the new experiments should be elaborated there. Is there a perspective potential future applications or extensions (multiple abundant spins, hyper-polarized systems)?
*The conclusions section has been rewritten it is much more extended now, discussing different aspects of the work.*

**5.** In the abstract, the sentence "In these schemes, the dipolar field can be decomposed into two components, each at the helm of its own transfer." is obscure in its meaning.
*This sentence has replaced (see answer to comment 1 of Malcolm Levitt)*

**6.** p.2.l.34: "PFG" is maybe not suitable in this context, "field gradient" might be appropriate here, but are there any assumptions about the field gradient (it appears a linear field gradient is assumed n Eq. 1 and throughout the paper)
*PFG has been changed to field gradient in this sentence. I believe the gradients need not necessarily be linear for the theory to be valid. In the simulations, the gradients have been assumed to be linear. This is now clearly stated. If necessary, the simulation program could be adapted to take non-linearity of gradients into account.*

**7.** p.2.l.36: "and S" the S spins are not involved at that point
*The gyro-magnetic ratio in this equation is the one of the spins that feel the dipolar field. I prefer to keep both A and S in the phrase since it stresses the fact that we are discussing a homo-nuclear spin system and that the following equation is applicable to both. In the revised version $\omega_D$ is not anymore defined as "the amplitude of the dipolar field" since that is not strictly true and only leads to misunderstandings.*

**8.** p.2. Eq. (2) As the usage of symbols deviates from the cited literature, it might be in order to discuss the relation to previous work more thoroughly. The choice of the symbol ω for both the offset and the rf field strength, while justifiable, is not increasing readability.

*The symbol $B_d$ has been introduced which is predominantly used in previous works ($\omega_D$ has been changed into $\omega_d$ to be consistent). Also it has been specified that in much of previous work the rf field did not need to be taken into account. I prefer to keep ω.*

**9.** p.2. l.47: the bold symbol $\mathbf{m}^S$ is not formally introduced, apparently a vector of magnetization (but is it normalized as well?).

*$m^S$ and $m^A$ are now defined.*

**10.** p.3. li.54: eq. A6 should be moved to the main text for reading convenience as it is referred to frequently.

*This equation has been put into the main text (I also kept it in the appendix, even if it is redundant).*

**11.** p.3.li55: how (using which program? Probably the one in the supplement, but it isn't mentioned in the main text) were the simulations made, information for reproducing those simulations should be included

*This calculation was done with the old code described previously, the reference has been added. The dashed lines in fig 1d have been calculated with the program in the supporting information. This has now been mentioned explicitly in the text.*

**12.** p.3.: Results using WALTZ-16 are mentioned but no data are shown.

*Data are not shown because the result of the approximate solution described in this work gives exactly the same (at least visual) result as the exact numerical calculation (as is also the case for continuous non-modulated irradiation). The results of both types of simulations are identical to the dashed curves in figure 1 for these irradiation schemes. This is now specified in the caption of this figure.*

**13.** p.4.li72:"more laborious simulations" – simulations should be described more specifically.

*A reference to the article where the simulations are described has been added to the sentence.*

**14.** p.4: In Section 3 Experiments, a clear structure is missing. I suggest to move the more theoretical discussion up to li. 96 into the Theory Section, and start Section 3 with an overview of tall he experiments used in the paper.

*These suggestions have been implemented.*

**15.** p.4: A more fundamental issue occurs with eq. (4), considering that eq. (3) clearly allows for terms with more than two spin operators. What justifies the restriction to two-spin operators?

*In the paragraph below new eq. 5, it is now explained why we only need two spin operators in this work (with a reference to Lee et al.). See also answer to comment 3 of Malcolm Levitt.*

**16.** p.4.li94: "timescale in which RD occurs can be more than an order 95 of magnitude shorter" please provide a reference/justification

*A reference has been added to a review article that discusses both phenomena*

**17.** p.5.Fig.2: The coherence pathway shows the selection of only one coherence during the evolution time. TPPI is designed to allow for +1 and -1 coherence orders during t1, which contradicts the gradient selection scheme shown. This point requires clarification. Does the mixing sequence really only convert -1 to +1 coherence orders? One would assume there must be parallel pathways.
*I thank the reviewer for pointing this out. In fact, TPPI is superfluous. For the spectra in the revised version, the data have been reprocessed with half of the time increments (removing the odd increments). The mixing sequence convert almost an infinite amount of coherence orders, but only the -1 to +1 is selected (considering that we detect the -1 coherence). Usually, one would expect phase twists in such a spectrum but due to the inhomogeneous line-shapes these cancel out (see new reference to Ernst et al.). This is also discussed in the supporting information (new figure SI3). In the experimental section I added details of the processing of the 2D.*

**18.** The choice of using indices 1,2,3 for the orthogonal gradients instead of x,y,z is unusual.
*This choice is needed for this work, since often the direction is varied. Since it is unusual, I specified it in figure caption 2.*

**19.** p.5.li.100: Does watergate really ensure in-phase spectra?
*The statement was indeed incorrect. It does not ensure in-phase spectra, it refocuses the chemical shift evolution which is now written in the text.*

**20.** p.6: Is the concept of a sliding window in the t2 dimension new?
*It is closely related to the "chunk selection" in pure shift experiments. A reference has been added.*

**21.** p.6: The benefits of the spectra obtained by this new pulse sequence are shown but not discussed. It should be clearly pointed out what novel advantages are achieved, also for a non-expert reader, and which limitations exist. Why were particular sections chosen in Fig. 3?
*The benefits and limitations are now discussed at the end of the section. The sections chosen were the most crowded part and the one with the weakest and most complex signal due to scalar coupling pattern. This is now mentioned in the text.*

**22.** p.7: Section 3.2
Some introductory sentences justifying the design of the new pulse sequences might be in order. In this reviewer's opinion, this is the most interesting aspect of this paper. Maybe state first, what the pulse sequences have been designed for.
*Some introductory sentences have been added to the beginning of the section, drawing the parallel to the original CRAZED sequence.*

**23.** The first § of this section contains a logical jump. "The commutator of the lowest order term" what does this refer to? Maybe the sentence just needs to be rearranged to have eq. 4 upfront?
*The sentence has been rearranged.*

**23.** Definitions seem to be missing for **r,** p, q
*These symbols are now defined*

**25.** p.8: Fig. 4. The presentation of the pulse sequences (actually all pulse sequences in the paper) should be improved by including delay labels and following more usual visual clues for 90° and 180° pulses (narrow and wide rectangles). The exceptional 0 –2 coherence transfer should be commented on, emphasizing its occurrence being due to the DF effects.

*The presentation of the pulse sequences has been improved. Including clearer differentiation of 90° and 180° pulses and shaped pulses. The delays are also drawn in the sequence. The fact that the DF induces a change in coherence order is now stressed in several parts of the manuscript, especially since this is not possible if it acts only during free precession delays.*

**26.** p.9: Fig.5 contains a plethora of information on angular and mixing time dependence, which should be discussed in more detail.

*The figure, in particular 5a and 5b are discussed more in detail.*

**27.** p.9. Why was a methyl signal chosen for the investigations? Couldn't the known "anomalies" of CH3 groups have an impact?

*The methyl groups have been chosen since it is an intense signal, it is furthest away from the solvent signal, it is a singlet signal and has long relaxation times. It is easy to quantify even in sequences without solvent suppression. There is no evidence of any anomaly, the build-up seems not to be fundamentally different compared to the other signals (see for example figure 3). In our previous work, where we only looked at this signal, the signal intensities could easily by fitted without invoking any special theory.*

**28.** Similar to the comments on the previous section, first the purpose or motivation for the new experiments should be outlined. The experiments are highly complex and the explanation, while detailed in describing, what the elements of the pulse sequence are doing, the goal seem elusive. There is a lot of potentially useful insight in this subchapter, but the overall presentation is a bit confusing. Some more systematic structuring would be useful.
The text would also benefit from language editing in particular in this section.

*Some introductory sentences are added. I think that separating the pulse sequences in the figure and related changes in the text makes it more structured and less confusing. I tried to correct the language over the entire article.*

**29.** p.10 Fig. 6: It maybe worthwhile to draw the coherence transfer diagram with separate sub-pathways for the rare and the abundant spin, similar to the way in which pathways in heteronuclear pulse sequences are often illustrated.

*Thanks for the suggestion. I think that by separating the figures it is now much clearer and there is no need to separate the pathways. However, I followed this suggestion for fig. 4b.*

**30.** p.10 eq.8: Where does this eq. come from?

*It has now been specified that one has to discard the parts of the density operator that has been de-phased by $G_c$.*

**31.** p.11: The decomposition outlined is highly enlightening, it might be more instructive to include some transfer diagrams connecting operator terms by arrows indicating the transfers enabled by the DF, with the DF represented as a (pseudo) propagator.

*I kept the decomposition description as is. I did not find a satisfactory diagram representation to describe the experiment.*

**32.** p.10-13: I believe a clearer designation of the pulse sequence variants can simplify discussion. Thus overly wordy expressions like "except that the PFG Ga which was placed just before acquisition has been moved in front of the DIPSI-2 pulse-train" can be avoided. Maybe additional panels in Fig. 6 may be a solution.
*I followed this suggestion, figure 6 now contains 4 panels.*

**33** I have put some annotations into the pdf-file I am sending separately, which include a few suggestions for fixing language issues.
*Thanks a lot for the annotations in the pdf. They have been taken into account in the revised manuscript.*

**34.** Figure 1: axis labelling and annotations font sizes should be improved. What is the frequency offset between A and S in (d)?
*The figure is now in a double column format which increases de readability. The frequency difference is indicated in the caption.*

**35.** The pulse sequence diagrams would benefit a lot from improvements wrt. to label positions and sizes and a more conventional representation of 90, 180° pulses by narrow and wide rectangles and for the indication of selective pulses.
*The sequences are now clearer with delays, better differences between 90° and 180° pulses and shapes for the selective pulses.*

**36.** Figure 2: The coherence pathway has been commented on above. The spectrum is too small and needs to be annotated to be meaningful (peak assignments). The sample composition could be stated in the caption rather than the main text. The positions of the 13C satellites should be marked. The watergate delays are not given.
*The spectrum is enlarged. The sample composition is mentioned in the caption. The full spectrum with all peak assignments is put in the supporting information. The satellites are marked with asterisks. The delays are now drawn in all sequences.*

**37.** The quality (size, graphics resolution) and the labelling of the spectra needs to be improved.
*Most figures are double column format now, which improves the readability and the labeling has improved.*

**38.** Equation 5 formally is not an "equation".
*The window function has now been defined as an equation (eq. 6 in the revised version).*

**Tom Barbara**
Having followed the "distant dipolar field" topic from the beginning, I can empathize with Malcolm's comments on what the exact conditions are for the validity of the main equations of motion. The literature always goes back to Deville et. al. and I could never find a paper on a full derivation with details. Faced with that I attempted to work it out for myself and I offer my old

notes on the topic for those that may be interested. As Malcolm points out, the general dynamics is non local in r space. That locality can be used is a result of the assumption that the magnetization is purely harmonic in space and therefore in k space we get a Dirac function in (k-K) where the capitol K is the assumed harmonic variation, usually achieved by gradient modulation followed by storage along the z axis. However, these expressions always offered also pertain only to cylindrical symmetric distributions of magnetization. The rotational transformation problem in full is actually very complicated from what I can see, and back in the day I tried to attack it using vector spherical harmonics but I must admit that I lost interest in the problem, since it can become tedious to use all that angular momentum machinery.

I believe that readers will want to know some of these things and I feel that the paper can be improved by not just parachuting into the final equations, but giving (briefly!) some of the details. For example, what is the field from the magnetization and then what consequences do these have for the Bloch dynamics. This then invites a reader to continue, rather them forcing the reader to work it out on their own and stopping the flow of thought.

*Thanks a lot for your insights and the copy of your notes, which can serve the interested reader as a starting point for further exploration. I prefer not to delve further into the theory. I tried to keep the theory and the equations to a minimum that is needed to understand the simulations and the pulse sequence design described in the paper.*